# Fat: Quality, or Quantity? What Matters Most for the Progression of Metabolic Associated Fatty Liver Disease (MAFLD)

**DOI:** 10.3390/biomedicines9101289

**Published:** 2021-09-22

**Authors:** Olga Estévez-Vázquez, Raquel Benedé-Ubieto, Feifei Guo, Beatriz Gómez-Santos, Patricia Aspichueta, Johanna Reissing, Tony Bruns, Carlos Sanz-García, Svenja Sydor, Lars P. Bechmann, Eva Maranillo, José Ramón Sañudo, María Teresa Vázquez, Arantza Lamas-Paz, Laura Morán, Marina S. Mazariegos, Andreea Ciudin, Juan M. Pericàs, María Isabel Peligros, Javier Vaquero, Eduardo Martínez-Naves, Christian Liedtke, José R. Regueiro, Christian Trautwein, Rafael Bañares, Francisco Javier Cubero, Yulia A. Nevzorova

**Affiliations:** 1Department of Physiology, Genetics and Microbiology, Faculty of Biology, Complutense University of Madrid, 28040 Madrid, Spain; olgaeste@ucm.es (O.E.-V.); rabenede@ucm.es (R.B.-U.); 2Department of Immunology, Ophthalmology and ENT, School of Medicine, Complutense University of Madrid, 28040 Madrid, Spain; feirguo@163.com (F.G.); csanz17@ucm.es (C.S.-G.); arlamas@ucm.es (A.L.-P.); lmoran@ucm.es (L.M.); mamazari@ucm.es (M.S.M.); emnaves@ucm.es (E.M.-N.); regueiro@ucm.es (J.R.R.); rbanares@ucm.es (R.B.); fcubero@ucm.es (F.J.C.); 3Department of Physiology, Faculty of Medicine and Nursing, University of Basque Country UPV/EHU, 48940 Leioa, Spain; bgomezsantos@gmail.com (B.G.-S.); patricia.aspichueta@ehu.eus (P.A.); 4Biocruces Health Research Institute, 48903 Barakaldo, Spain; 5Centro de Investigación Biomédica en Red de Enfermedades Hepáticas y Digestivas (CIBEREHD), Instituto de Salud Carlos III, 28220 Madrid, Spain; pericasjm@gmail.com (J.M.P.); javiervaq@hotmail.com (J.V.); 6Department of Internal Medicine III, University Hospital RWTH Aachen, 52074 Aachen, Germany; joreissing@ukaachen.de (J.R.); tbruns@ukaachen.de (T.B.); cliedtke@ukaachen.de (C.L.); ctrautwein@ukaachen.de (C.T.); 7Department of Internal Medicine, University Hospital Knappschaftskrankenhaus, Ruhr-University Bochum, 44801 Bochum, Germany; svenja.sydor@ruhr-uni-bochum.de (S.S.); lars.bechmann@rub.de (L.P.B.); 8Department of Human Anatomy and Embryology, School of Medicine, Complutense University of Madrid, 28040 Madrid, Spain; evamaranillo@med.ucm.es (E.M.); jrsanudo@med.ucm.es (J.R.S.); tvazquez@med.ucm.es (M.T.V.); 9Instituto de Investigación Sanitaria Gregorio Marañón (IiSGM), 28009 Madrid, Spain; 10Endocrinology Department, Vall d’Hebron University Hospital, Vall d’Hebron Institute for Research (VHIR), 08035 Barcelona, Spain; aciudin@vhebron.net; 11Liver Unit, Internal Medicine Department, Vall d’Hebron University Hospital, Vall d’Hebron Institute for Research (VHIR), 08035 Barcelona, Spain; 12Servicio de Anatomía Patológica, Hospital General Universitario Gregorio Marañón, 28007 Madrid, Spain; isabel.peligros@salud.madrid.org; 13Servicio de Aparato Digestivo, Hospital General Universitario Gregorio Marañón, 28007 Madrid, Spain; 1412 de Octubre Health Research Institute (imas12), 28041 Madrid, Spain

**Keywords:** obesity, metabolic associated fatty liver disease (MAFLD), steatohepatitis, fibrosis, palmitic acid (PA)

## Abstract

Objectives: Lately, many countries have restricted or even banned transfat, and palm oil has become a preferred replacement for food manufacturers. Whether palm oil is potentially an unhealthy food mainly due to its high content of saturated Palmitic Acid (PA) is a matter of debate. The aim of this study was to test whether qualitative aspects of diet such as levels of PA and the fat source are risk factors for Metabolic Syndrome (MS) and Metabolic Associated Fatty Liver Disease (MAFLD). Methods: C57BL/6 male mice were fed for 14 weeks with three types of Western diet (WD): 1. LP-WD—low concentration of PA (main fat source—corn and soybean oils); 2. HP-WD—high concentration of PA (main fat source—palm oil); 3. HP-Trans-WD—high concentration of PA (mainly transfat). Results: All types of WD caused weight gain, adipocyte enlargement, hepatomegaly, lipid metabolism alterations, and steatohepatitis. Feeding with HP diets led to more prominent obesity, hypercholesterolemia, stronger hepatic injury, and fibrosis. Only the feeding with HP-Trans-WD resulted in glucose intolerance and elevation of serum transaminases. Brief withdrawal of WDs reversed MS and signs of MAFLD. However, mild hepatic inflammation was still detectable in HP groups. Conclusions: HP and HP-Trans-WD play a crucial role in the genesis of MS and MAFLD.

## 1. Introduction

Metabolic Associated Fatty Liver Disease (MAFLD) is characterized by the presence of hepatic steatosis in addition to any of the following three criteria: overweight or obesity, presence of Type 2 Diabetes Mellitus (T2DM) or evidence of metabolic dysregulation [1]. Since its initial description [2] in 1980, the incidence and prevalence of MAFLD has increased dramatically, in parallel with the global epidemic of obesity. Today, it is the most widespread liver disease in human history, affecting about a quarter of the global population. In European countries, 25% of individuals aged ≥15 years experienced MAFLD in 2015 [3]. Alongside the progressive liver damage, MAFLD is becoming an established risk factor for the leading causes of death and disability, namely cardiovascular disease and cancer. Despite these already staggering facts, the MAFLD burden is expected to grow in the coming decades, compromising individual health and health-care systems and causing substantial economic and well-being losses [4,5].

MAFLD is often referred to as a self-inflicted disease, implying the influence of personal behavioral and dietary choices. However, the impact of the surrounding “fast and processed food” obesogenic environment on the choices of children and adults should not be neglected [1]. Processed or ready-to-consume food products are characteristically energy-dense, fatty, sugary or salty. At the same time, they are hyper-palatable, cheap and attractive, therefore highly dominant, and they comprise the majority of the total energy intake among young adults currently [6,7].

The food industry and especially the fast-food industry, extensively uses the types of lipids with long shelf life that are easy and inexpensive to produce [8]. Processing natural liquid vegetable oils, by adding hydrogen, modifies their texture from liquid to solid forms and results in the development of transfats (or partially hydrogenated fats). Trans Fatty Acids (FA) give food a desirable taste and texture. The main examples of industrial Trans FA are margarines, commercially baked products, deep-fried fast foods, packaged snack foods and other prepared food [9]. According to the U.S. Food and Drug Administration (FDA), in 1994–1996 we consumed about 5.6 g of transfat per day [10].

In the early 1990s, researchers began to identify the adverse health effects and reported that intake of Trans FA increased the risk of coronary heart disease by altering the ratio between Low-Density Lipoprotein (LDL) cholesterol and the “good” High-Density Lipoprotein (HDL) cholesterol [11].

Increased consumer awareness of the health implications of Trans FA resulted in local and state efforts to limit or ban their use [9].

“Anti transfat” measures introduced by governments accelerate the use of an accessible alternative—palm oil, obtained from the fruit of the palm tree (*Elaeis guineensis*). Palm oil is abundant, low-cost and chemically stable. Currently, it is the most widely produced edible vegetable oil in the world [12]. Palm oil is rich in saturated FAs, especially Palmitic Acid (PA), unlike most of the vegetable oils, which are rich in unsaturated fat. For this reason, palm oil is semisolid at room temperature, making it suitable for the formulation of processed food [13].

Recently, the impact of palm-oil consumption on the heart, especially in the development of coronary artery disease, was a matter of considerable debate. The main argument against the use of palm oil is the fact that it contains saturated PA, which may case elevation of total cholesterol and LDL levels [14]. 

Currently, transfat and PA consumption remain a major source of concern within health policies regarding the reduction of cardiovascular and metabolic-disease risk [15,16]. However, evidence of the consequences of transfat and PA consumption in the development of MAFLD is scarce and even controversial. In the present study, we investigated whether this type of FA could have an impact on MAFLD progression.

## 2. Materials and Methods

### 2.1. Animal Experimentation

All animal procedures were performed according to Spanish legal requirements and animal protection law, and were approved by the authority of environment conservation and consumer protection of the Regional Government of Madrid (PROEX210/18, 15 June 2018). All animals were maintained in the Animal Facility at the Faculty of Biology, Complutense University Madrid, in a temperature-controlled room with 12-h light/dark cycle with ad libitum feeding condition, according to the guidelines of the Federation for Laboratory Animal Science Associations (FELASA).

We randomly distributed 10-week C57BL/6J male mice into 4 groups with a sample size of 7 to 9 animals per group.

Control (CTRL) group was fed with chow diet and normal water. Three treated groups received different types of Western Diet (WD) (Appendix A) and sweetened drinking water containing 6.75% D-glucose (Sigma-Aldrich, St. Louis, MO, USA). For the withdrawal study, we established a model in which mice were exposed to WD and sweetened drinking water for 14 weeks, followed by 20 days of chow diet and normal water. The corresponding controls received chow diet and normal water for 17 weeks.

### 2.2. Supplementary Material and Methods

Please see Appendix A and Appendix A for further information regarding material and methods on Animal Experimentation, Glucose Tolerance Test (GTT), Histological analysis, Immunofluorescence (IF) and Immunohistochemistry (IHC) staining, Image analysis, RNA isolation and RT-qPCR, Western blot, and Lipid extraction and quantification.

### 2.3. Statistical Analysis

Data were expressed as mean ± SD. GraphPad Prism version 8.0 (https://www.graphpad.com/scientific-software/prism/), accessed 27 December 2018 (San Diego, CA, USA) was used for statistical analysis and graph design. Statistical significance was determined by one-way analysis of variance ANOVA, followed by a Tukey post hoc test. One-way paired ANOVA followed by Bonferroni’s post hoc test was used to evaluate the differences between feeding groups. Significant differences are denoted by * for *p* < 0.05, ** *p* < 0.01, *** *p* < 0.001; **** *p* < 0.0001.

## 3. Results

### 3.1. Western Diet (WD) Is Associated with the Development of Obesity and Metabolic Syndrome (MS) 

Ten week-old C57BL/6J male mice were fed with three different WDs and sweetened water for 14 weeks. The three types of WD had a similar content of fat (40%), fructose (22%) and cholesterol (2%) (Appendix A). However, the origin of fat and the level of PA was greatly different between the groups. In group 1, the level of PA was low (LP-WD), and the corn and soybean oils were the main source of fat. In group 2, the level of PA was high (HP-WD) due to the palm oil added to the formula. In group 3, the level of PA was also high (HP-Trans-WD), but the transfats were the main source of the fat (Appendix A). Glucose was added to the drinking water in order to potentiate fructose absorption from the diet [17]. Mice fed with the control chow diet and filtered tap water were used as controls.

Body Weight (BW) steadily increased in all treated groups during the experimental period. Nevertheless, as shown in (Figure 1A,B), throughout the 14-week period the mice fed with LP-WD, on average, gained less weight compared to HP-WD and HP-Trans-WD. 

Obesity is associated with significant changes in White Adipose Tissue (WAT), which has profound systemic and hepatic consequences [18]. Mice fed with all types of WD diet accumulated epididymal WAT (eWAT) and had higher visceral fat mass compared to control animals (Figure 1C, Appendix A). The administration of WDs also led to hypertrophy and increase of adipocyte size up to 80% after 14 weeks of WD feeding (Figure 1D, Appendix A). Additionally, Crown-Like Structures (CLS) formed by macrophages aggregated around dying adipocytes were observed in all WD groups (Figure 1D).

Basal glucose levels after 12 h of fasting, did not increase in all WD treated groups compared with CTRL (Figure 1E, left panel). However, a Glucose Tolerance Test (GTT) revealed significantly impaired glucose tolerance in mice fed a HP-Trans-WD (Figure 1E, right panel, Appendix A). 

Finally, the serum level of total cholesterol was significantly higher in HP-WD and HP-Trans-WD fed animals compared with LP-WD and CTRL groups (Figure 1F).

### 3.2. WD Consumption Triggered Hepatomegaly

Obesity, hypercholesterolemia and glucose intolerance are the main features of MS and are closely associated with the progression and severity of MAFLD [19]. Anatomopathological examination of livers revealed that the hepatic parenchyma of all mice fed with WD was significantly enlarged and pale yellow in color. Accordingly, the liver mass and hepatosomatic ratio were increased in all treated groups compared with CTRL mice (Figure 2A).

Hepatomegaly was caused by a combination of hypertrophy and hyperplasia. Hepatocytes of livers from animals of all WD-fed groups were almost twice enlarged compared to CTRLs, as shown by phalloidin staining (Figure 2B). At the same time, Ki-67 staining revealed mild but significantly higher cellular proliferation in all treated groups (Figure 2C).

Evaluation of histologic sections, by an experienced pathologist, revealed significant hepatic steatosis in 30–80% hepatocytes of WD-fed animals. The cellular content was a mixture of micro- and macrovesicular vacuoles, mainly located in the periportal regions (Figure 2D, Appendix A). These vacuoles were confirmed to contain lipids using Oil Red O (ORO) staining in liver cryosections (Figure 2E).

### 3.3. WD Feeding Induced Lipidome Alterations in Murine Livers

Next, a quantitative lipidomic analysis was performed in order to identify lipids discriminating the pathological statuses of the liver. A lipidomic approach was taken to quantify the major lipid classes.

Lipids were extracted from the liver tissue and further separated by Thin Layer Chromatography (TLC). Total hepatic Triglyceride (TG) content was markedly increased in all mice fed with WD compared with CTRL group (Figure 3A). However, the level of Diglycerides (DG) was significantly increased only in mice fed with HP-WD and HP-Trans-WD (Figure 3B). Importantly, hepatic Free Cholesterol (FC) levels significantly increased in both groups fed with a HP diet (Figure 3C). The total Phosphatidylcholine (PC) content was decreased (Figure 3D, left panel), whilst Phosphatidylethanolamine (PE) levels significantly increased (Figure 3D, middle panel) and the PC/PE ratio decreased only in mice fed with a transfat diet (Figure 3D, right panel). A trend towards decreased Phosphatidylserine (PS) levels was found in the HP-Trans-WD group (Figure 3E).

### 3.4. WD Altered the Balance between Fat Storage and Oxidation in the Liver

Subsequently, we assessed the effects of all three types of WD on major molecular mechanisms that regulate lipid metabolism in the liver. Consistently with previous studies [20], the excessive lipid load in all treated groups induced the expression of *Pparγ*, early induced lipogenic transcription factor mediating an adipogenic transformation of hepatocytes (Figure 4A, left panel).

We also found strong increase of *Scd1* mRNA expression in all treated animals (Figure 4A, right panel). Scd1 desaturase converts saturated FA to monounsaturated FA [21], the major substrate for TG in the liver [22]. Hence, the substantial increase in *Scd1* expression was mainly induced by the excessive dietary fat, as mRNA of main regulators of de novo lipogenesis *Acc* and *Srebp1* were not increase after feeding with WDs (Figure 4B). Furthermore, FASN protein expression was remarkably suppressed in HP-Trans-WD group (Figure 4C).

Increased lipid load in the liver resulted in enhanced lipid oxidation only in LP-WD, as observed by raised *Acox* and CPT-1c expression levels. In sharp contrast, the application of both diets with high levels of PA seems to not induce the transcription of genes and proteins for an adequate lipid oxidation (Figure 4D,E).

Finally, decreased *Apob* levels may indicate the impairment in Very Low Density Lipoprotein (VLDL) secretion in HP-Trans-WD (Figure 4F).

### 3.5. WD with High Levels of PA and Trans Fat Increased the Risk of MAFLD-Associated Hepatitis and Fibrosis

Excessive lipid accumulation drives to cell death and inflammation in the liver [23]. Plasma levels of Alanine Aminotransferase (ALT) and Aspartate Aminotransferase (AST), important clinical markers of hepatocellular injury were significantly elevated in animals fed with HP-Trans-WD (Figure 5A). Thus, the detection of hepatic cell death in situ revealed significantly higher numbers of TUNEL positive cells in HP-WD and HP-Trans-WD fed livers in comparison with LP-WD-fed animals and the controls (Figure 5B).

All WD treated groups showed an increased accumulation of F4/80 positive liver macrophages, as assessed by Immunofluorescence (IF) staining (Figure 5C). Additionally, the mRNA expression of *Tnf-α* was significantly increased only in mice fed either with a HP-WD or a HP-Trans-WD (Appendix A).

Cell death, inflammation and TNF-α overproduction induces the activation of Hepatic Stellate Cells (HSCs) in the liver [24]. Consistently, significant expression of α-Smooth Muscle Actin (α-SMA), a marker of HSCs activation, was detected by Immunohistochemistry (IHC) staining in both groups treated with HP-WD and HP-Trans-WD (Figure 5D). Activated HSCs are the major source of Extracellular Matrix (ECM) during progression of fibrosis [2]. Hence, Sirius Red (SR) staining clearly demonstrated that feeding with HP-WD and HP-Trans-WD induced collagen expression in the liver (Figure 5E, Appendix A). These findings were additionally confirmed by IF staining for Collagen I (Figure 5F).

### 3.6. The Withdrawal ofWD, Reversed MAFLD in All Treated Groups Independently of the Diet Composition

Lastly, we investigated whether WD withdrawal could ameliorate MS and MAFLD-related liver phenotype. The restoration of a chow diet feeding regimen for 20 days was performed after 14 weeks of WD in all treated groups, while CTRL group was maintained on chow diet for 17 weeks in total (Figure 6A). The switch from all different WDs to chow feeding diet, induced a decrease in body weight (Figure 6B, Appendix A) and significantly reduced hepatomegaly (Figure 6C, Appendix A), compared with the corresponding 14 weeks feeding period of the three different types of WDs that has been used, achieving similar levels of the CTRL group. The plasma levels of transaminases and cholesterol progressively decreased over the duration of withdrawal in all treated groups, and eventually reached almost the same level as in the CTRL animals (Figure 6D). Importantly, WD withdrawal normalized impaired glucose intolerance in mice fed with HP-Trans-WD (Figure 6E,F).

In light of these metabolic improvements, pathological changes in the liver were explored next. All WD withdrawal groups, showed significant improvement in hepatocellular morphology (Figure 7A) and attenuated hepatic steatosis with improved hepatic TG content (Figure 7B,C). Moreover, even after 3 weeks of diet withdrawal, we observed low-grade level of infiltration in both groups treated with high PA (Figure 7D). However, the withdrawal was capable to revert fibrotic changes in all treated groups (Figure 7E).

## 4. Discussion

Over the past two decades, some light has been shed on a number of aspects of MAFLD pathogenesis and the complex relationships between liver steatosis, obesity and modifiable risk factors such as food habits and sedentary lifestyle. Despite the increase in knowledge, there is still no universally approved medical treatment for MAFLD patients [4]. The current body of research suggests that the best way to prevent MAFLD is the combination of a balanced diet with regular exercise to achieve a healthy weight. In contrast, excessive fat consumption is strongly related with accumulation of fat in the liver. However, the quality of fat might be as important as the quantity for MAFLD progression. Although the effects of specific types of dietary fat on cardiovascular diseases have been more or less widely studied, there have been only a limited number of investigations examining the effects specifically on MAFLD in this regard. Consequently, it is critical to investigate the effects of transfat and HP oils as possible triggers to the development of MAFLD, since the consumption of ultra-processed foods, rich in both oils, is increasing worldwide.

Therefore, the aim of the present study was to investigate the effects of three different WDs on the liver and the pathophysiological modulation that these diets perpetrate as a mechanism to develop MAFLD. The consumption of all types of WDs rich in fat, fructose and cholesterol for 14 weeks predisposed mice to the development of obesity, hepatomegaly, hepatic cell death and steatosis. Even LP-WD promoted obesity, hepatomegaly and significant liver fat accumulation, with modulation of FA metabolism. Thus, only HP-WD and HP-Trans-WD induced hypercholesterolemia and immune cell infiltration resembling the pattern of Nonalcoholic Steatohepatitis (NASH). Moreover, only mice treated with HP-WD and HP-Trans-WD developed significant liver fibrosis, an important hepatic feature of MAFLD that is usually observed in patients with persistent necroinflammatory changes [25] (Appendix A).

Our observations are in line with previously published murine models fed with a high fat diet (HFD) containing transfats—mice receiving American Lifestyle-Induced Obesity Syndrome (ALIOS) diet [26]. ALIOS mice became obese and developed severe hepatic steatosis with associated necroinflammatory changes. Plasma ALT levels increased, as liver TNF-α and procollagen mRNA, indicating an inflammatory and profibrogenic response to injury [25]. Feeding with a high-fructose medium-chain-transfat diet in another study was also associated with obesity, increased hepatic oxidative stress and a steatohepatitis-like phenotype with significant fibrosis [27].

The mechanisms of hepatic steatosis caused by transfats are the subject of considerable debate. We examined mRNA expression of key lipid metabolic genes involved in FA uptake, export and oxidation. Our data showed that the diet rich in transfat promotes hepatic lipid accumulation by more than one mechanism. We found that HP-Trans-WD in the liver reduced expression of *Apob* (hepatic TG secretion) and *Acox* (FA β-oxidation). Both decreased FA oxidation and secretion are not able to offset the diet-induced increase in intrahepatic lipid and together contribute to the diet-induced fatty liver and hepatic accumulation of fat.

Consistently, it has been demonstrated in vitro [28,29], as well as in human studies [30], that Trans FAs alter secretion and size of Apo-B100 containing particles produced by hepatic parenchymal cells and reduce the expression of FA Oxidation (FAO) enzymes [31]. In line with these observations, an increase of SCD1 activity in HP-Trans-WD was previously reported in obese subjects and associated with lower FA oxidation and higher fat storage [21].

Interestingly, we found that feeding with HP-Trans-WD did not increase the expression of key genes of de novo lipogenesis, but in fact FASN expression was profoundly suppressed. This appears paradoxical because FASN suppression markedly improves steatosis in other experimental models of fatty liver, and FASN antagonists are under development as plausible treatments for hepatic steatosis [32]. Therefore, it seems that in HP-Trans-WD fed animals, FASN suppression is likely a compensatory mechanism designed to reduce lipid synthesis under conditions in which lipid oxidation is reduced and TG export is inactive. Moreover, FASN expression is impaired by hepatic inflammation in experimental models characterized by severe hepatocellular damage and inflammation, as well as in patients with steatohepatitis [33]. Consistently, mice fed with HP-Trans-WD showed high levels of plasma markers of liver injury and significant inflammation in liver tissue, accompanied by increased *Tnf-α* expression. Concomitant with our observation, recent human studies showed that Trans FAs modulate human macrophage response, increasing the production of TNF-α [34].

Lipidomic analysis revealed a marked step from DG to TG (precursor/product) in HP-Trans-WD. These findings are highly consistent with human lipidomic MAFLD data published by A.J. Sanyal’s group [35]. Moreover, knockdown of Diacylglycerol Acyl Transferase (DGAT) ameliorates the fatty liver in the ob/ob mouse [36], suggesting that DGAT plays an important role in the development of hepatic steatosis in mice fed HP-Trans-WD. Several publications also supported the link between altered DG levels and insulin resistance [37].

DG are derived from lipogenesis and membrane phospholipids (PL). The lower level of de novo lipogenesis and the parallel decrease in PC suggest that the membrane PC may definitely contribute to the observed increase in DG in HP-Trans-WD mice in the current study [34]. Indeed, PC is one of the most abundant PL in mammals and a major component of cellular membrane lipids. Moreover, PC levels were reported to be decreased in the liver samples of patients with MAFLD [35,38]. In hepatocytes, up to 30% of PC comes from the conversion of Phosphatidylethanolamine (PE) to PC by Phosphatidylethanolamine N-Methyltransferase (PEMT). Hence, the significant decrease of the hepatic PC/PE ratio was previously observed in MAFLD patients [39]. Additionally, a loss-of-function polymorphism in the PEMT gene seems to be associated with susceptibility in MAFLD [39].

Furthermore, PC is the only phospholipid molecule that is known to regulate the assembly and secretion of lipoproteins. It has been found that low hepatic PC levels due to its synthesis disruption impair the VLDL secretion, and significantly decrease the levels of circulating VLDL lipoproteins and result in hepatic accumulation of TG [40], which is absolutely in line with low *Apo-B100* gene expression in HP-Trans-WD.

Additionally, a low PC/PE ratio possibly leads to a rearrangement of outer monolayer, loss of membrane integrity and increased permeability to pro-inflammatory molecules such as cytokines, initiating the inflammation in HP-Trans-WD mice [41].

Glycerophospholipids are another component of cellular membrane associated with cellular signaling and cellular apoptosis. Decreased levels of PS and PI were previously reported in liver biopsy samples of patients with NASH [38].

Another striking observation was the increase in FC in mice fed HP-Trans-WD. FC is well-known to be highly cytotoxic [42]. FC accumulation leads to liver injury through the activation of signaling pathways in Kupffer Cells (KCs), HSCs and hepatocytes, promoting inflammation and fibrogenesis [43]. Accordingly, we found significant hepatic infiltration of F4/80 positive cells as well as HSC activation in the HP-Trans-WD group. Concomitantly, Trans FA also increases the cellular accumulation and the secretion of free cholesterol by hepatocytes in vitro [30,40].

Notably, we found that only HP-Trans-WD induced a significantly impaired glucose tolerance after 14 weeks of feeding. Actually, transfats already have attracted critical attention as a potential modifiable risk factor for T2DM in several previous studies. Diet enriched in transfat intake has been associated with diabetes, insulin sensitivity and systemic inflammation [44,45]. Animal studies demonstrated that a high transfat diet causes weight gain and impaired insulin sensitivity in mice and monkeys [46,47].

Overall, our findings raise a red flag on the food high in transfats because of the harmful effects on liver and metabolism in general. However, the proposed industrial substitute—palm oil does not seem to be a good alternative either, mainly because of high saturated FA, especially PA (C16:0 ≈ 44%). PA treatment has been known to induce inflammation and cellular injury in various tissues. For example, Chen et al. showed that for each kilogram of palm oil consumed annually per capita, the mortality due to ischemic cardiomyopathy increases to the equivalent of 68 for 100,000 deaths in developed countries [48].

The effects of saturated PA on the liver are another pressing issue [8,16]. Boland et al. showed that the substitution of transfat with palm oil (GAN diet) resulted in maintained NASH phenotype in both ob/ob and C57BL/6J mice [49]. In our study, we found that animals fed with HP-WD develop obesity, hepatomegaly, steatosis, inflammatory cells infiltration in the liver and fibrosis as HP-Trans-WD. The changes in the lipid metabolism and lipidomic were also very similar between both groups: high Scd and Pparγ mRNA expression levels in combination with low FA oxidation (CPT1, Acox), leading altogether to significant fat accumulation in the liver. Moreover, the levels of TG, DG and FC were similarly increased in HP-WD.

The main difference between groups was that HP-WD feeding was not associated with glucose intolerance and did not prompt the increase of transaminases as HP-Trans-WD. In addition, the hepatic, metabolic and lipidomic changes were much more pronounced in HP-WD compared with LP-WD fed animals. In accordance, a previous in vitro study showed that saturated PA fatty acids induce hepatocyte lipoapoptosis, and it is more toxic than with unsaturated fatty acids [50].

Lifestyle modifications including diet, exercise and weight loss have been found to be effective in controlling MAFLD. Indeed, we found that WD withdrawal for only 3 weeks resulted in significant improvement in several features of MS and MAFLD, and in general was very efficient in returning most of the analyzed parameters to control-like levels. The withdrawal from all types of WDs promoted lower body mass; decreased visceral fat accumulation; remarkably improved steatosis, serum transaminases, cholesterol and hepatic TG content; and attenuated hepatic fibrosis. Importantly, it also normalized glucose tolerance in HP-Trans-WD. However, the specific lipid composition of the diet seems to also be important in this setting. Liver inflammation in HP and HP-Trans-WD as seen post-withdrawal, is not fully reversible. This, of course, raises concerns, pondering the frequent cycling of diet and weight in the population with obesity.

Mouse models have been extensively used in many mechanistic studies of MAFLD pathogenesis. In spite of this, the natural rodent diet is not fatty-acid-based, limiting the extrapolation of results to humans. Moreover, mice exhibit significant differences in lipid metabolism versus humans. For example, mice carry most of the plasma cholesterol in HDL, while humans carry much of it in LDL [51]. It is still reasonable to assume that many of the FA harmful effects for mice may also carry over to humans, both in terms of MS and MAFLD. Thus, careful validation and functional comparisons of lipidomic profiles must still be performed.

In summary, dietary recommendations to prevent and manage MAFLD should focus not only on quantity but also quality of the diet. Our study clearly demonstrated that modifying types of fat in the diet can lower risk of MAFLD progression and improve features of MS. Yet, it is very important to understand that “transfat free” and/or “palm oil free” labeling should not indulge people in buying high fat or processed food.

Altogether, this study provides important information on the role of different types of fat for development and progression of MAFLD and could be used to inform policy makers discussing the new European Union (EU) regulations to promote better food environments.

## Figures and Tables

**Figure 1 biomedicines-09-01289-f001:**
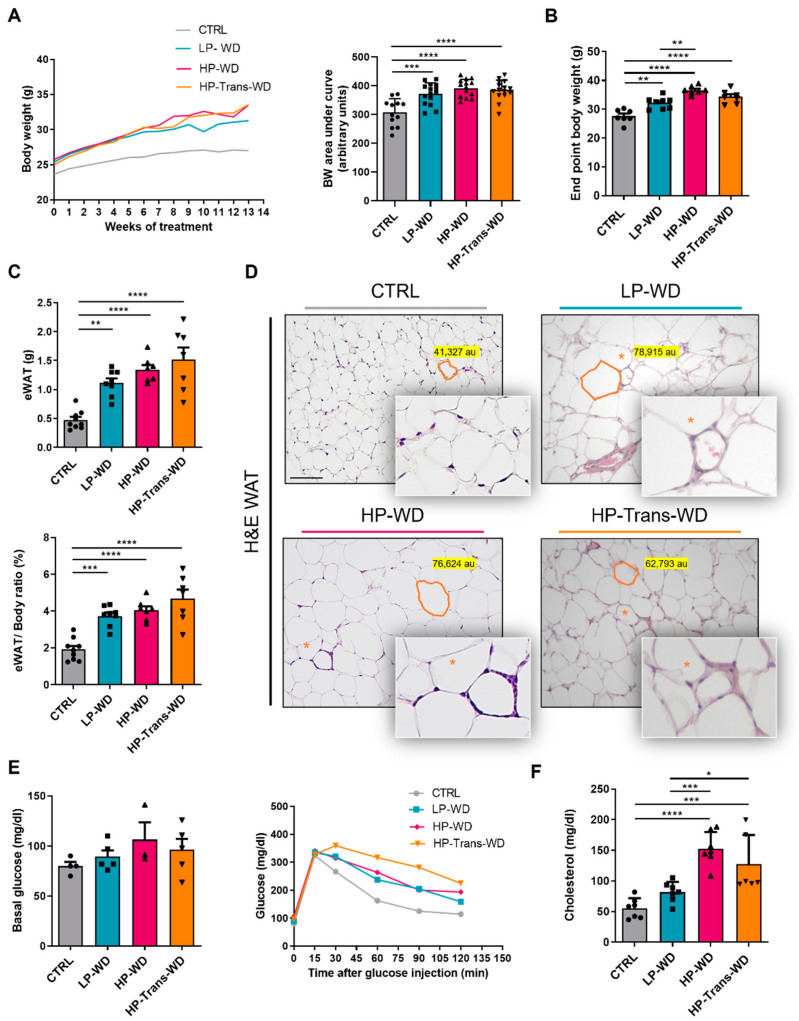
Metabolic profile of mice treated with different types of WD and control (CTRL) group. (**A**) Left: BW curve during the feeding period. Right: Area under the BW curve (arbitrary units) after 14 weeks. (**B**) End point body weight after 14 weeks of feeding (n = 7–9). (**C**) Upper: eWAT weight (g) (n = 7–9). Lower: eWAT weight–to-BW ratio (%) (n = 7–9). (**D**) Representative eWAT H and E. Asterisks indicate CLS. Scale bar = 100 μm (n = 4–5). (**E**) Left: Basal glucose levels in blood after 12 h of fasting (n = 3–5). Right: GTT curve after 14 weeks feeding. (**F**) Levels of cholesterol in serum (n = 6–8). * for *p* < 0.05, ** *p* < 0.01, *** *p* < 0.001; **** *p* < 0.0001.

**Figure 2 biomedicines-09-01289-f002:**
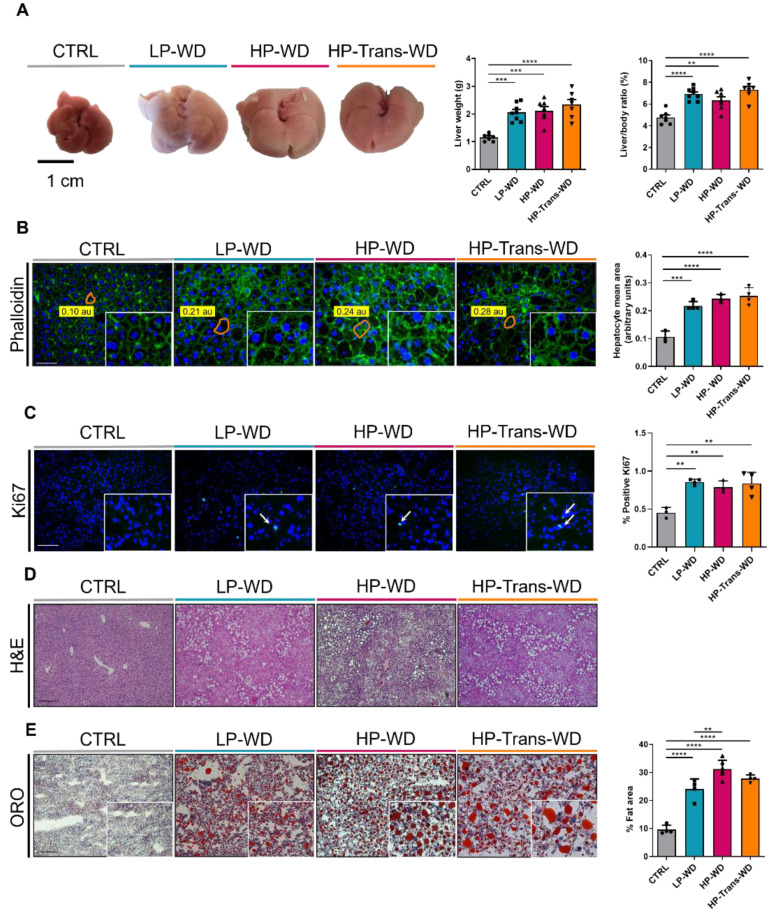
WD feeding leads to hepatomegaly and steatosis. (**A**) Left: Liver macroscopic images after 14 weeks of feeding. Middle: Liver weight (g) (n = 7–8). Right: Liver weight–to-BW ratio (%) (n = 7–8). (**B**) Representative phalloidin-stained liver images and size of hepatocytes in phalloidin-stained liver pictures quantified by ImageJ software. Scale bar = 100 μm (n = 3–4). (**C**) Ki-67 liver IF staining after 14 weeks of feeding. Positive proliferating cells are stained in green and indicated by arrows. Nuclei are stained in blue using DAPI as a counterstain. Scale bar = 100 μm (n = 3–4). (**D**) H and representative images. Scale bar = 100 μm. (n= 7–9). (**E**) Illustrative ORO-stained liver sections. Scale bar = 100 μm. Quantification of ORO-stained area (n = 4–6). ** *p* < 0.01, *** *p* < 0.001; **** *p* < 0.0001.

**Figure 3 biomedicines-09-01289-f003:**
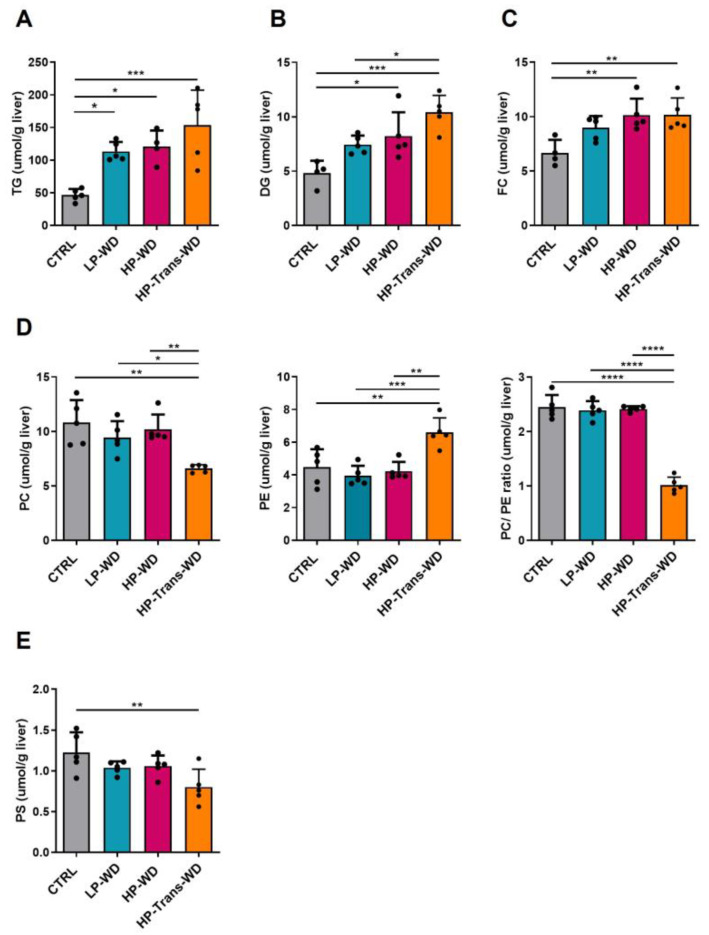
Hepatic lipidomic analysis in mice treated with WD and corresponding controls. Distribution of lipid classes within hepatic lipids (µmol/g of tissue): (**A**) Triglycerides (TG). (**B**) Diglycerides (DG). (**C**) Free cholesterol (FC). (**D**) Hepatic phospholipids Left: Phosphatidylcholine (PC). Middle: Phosphatidylethanolamine (PE). Right: PC/PE ratio. (**E**) Phosphatidylserine (PS) (n = 4–5). * for *p* < 0.05, ** *p* < 0.01, *** *p* < 0.001; **** *p* < 0.0001.

**Figure 4 biomedicines-09-01289-f004:**
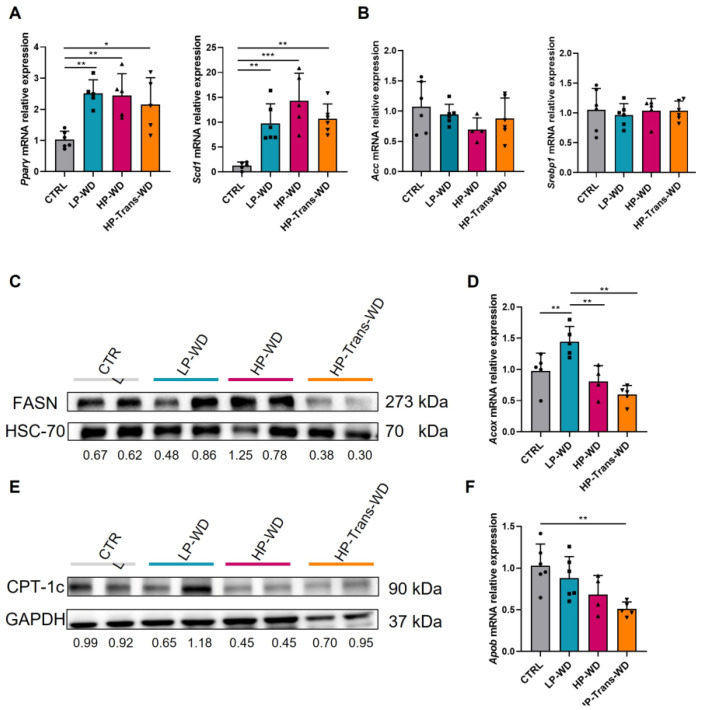
Changes in lipid metabolism in mice fed with WD. (**A**) *Pparγ, Scd1*, (**B**) *Acc, Srebp1* mRNA relative expression to GAPDH after 14 weeks on WD (n = 4–6). (**C**) FASN western blot using HSC-70 as a loading control. Ratio: normalization of FASN expression by densitometry. (**D**) Respective *Acox* mRNA expression relative to Gapdh (n = 4–6). (**E**) CPT-1c WB using GAPDH as loading control. Quantification of Western blot was performed by densitometry using Image J software. (**F**) *Apob* mRNA expression relative to Gapdh (n = 4–6). * for *p* < 0.05, ** *p* < 0.01, *** *p* < 0.001.

**Figure 5 biomedicines-09-01289-f005:**
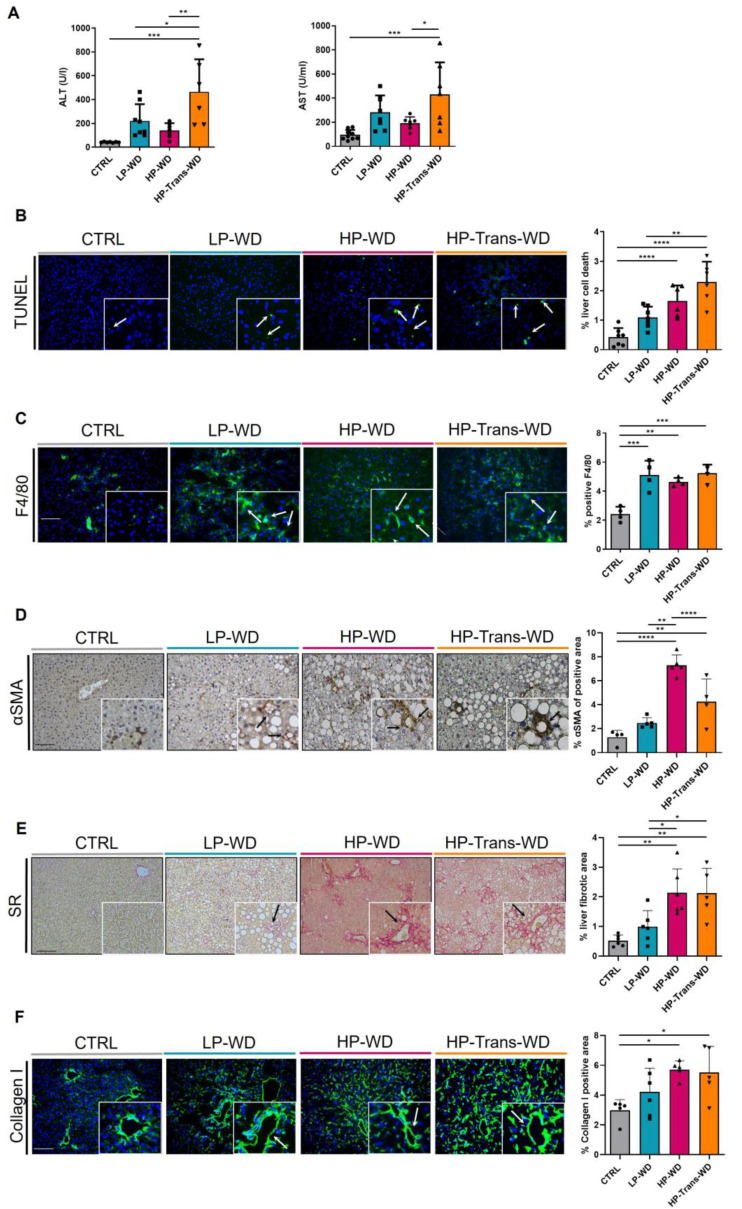
Hepatitis and hepatic fibrosis in mice on the WD for 14 weeks. (**A**) ALT and AST measurements in serum after 12 h of fasting (n = 7–9). (**B**) Representative TUNEL-stained photomicrographs at 14 weeks. Scale bar = 100 μm. Quantification of % TUNEL positive cells (n = 6–7). (**C**) F4/80 IF staining in liver sections of mice fed for 14 weeks. Positive immune cells are stained in green. Nuclei are stained in blue using DAPI as a counterstain. Arrows indicate F4/80 positive cells, respectively. Scale = 100 μm. Quantification of % F4/80 positive cells, using ImageJ software (n = 4). (**D**–**F**) Fibrosis-related stainings in liver and corresponding quantification of positive stained areas after 14 weeks of treatment. Representative liver images stained with α-SMA (IHC) (**D**), SR (**E**), Collagen I (IF) (**F**). Scale bar = 100 μm. (n = 4–7). * for *p* < 0.05, ** *p* < 0.01, *** *p* < 0.001; **** *p* < 0.0001.

**Figure 6 biomedicines-09-01289-f006:**
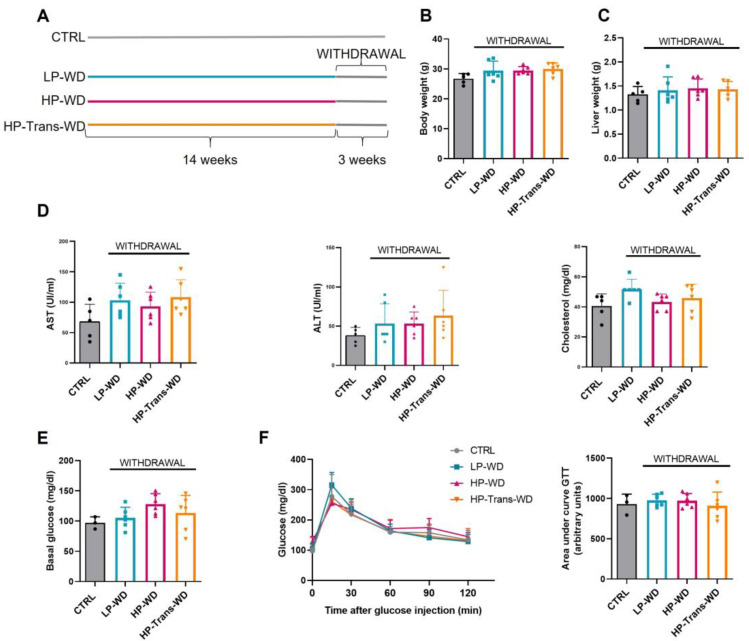
Effects of WD withdrawal on MS. (**A**) Mice were fed for 14 weeks with WD followed by 3 weeks of withdrawing. Effects of WD withdrawal: (**B**) Body and (**C**) Liver weight (n = 5–6). (**D**) AST, ALT, Cholesterol in serum after 12 h of fasting (n = 3–5). (**E**) Basal glucose levels in blood after 12 h of fasting. (**F**) Left: GTT curve after the period of withdrawal. Right: Area under the curve GTT (arbitrary units) (n = 3–6).

**Figure 7 biomedicines-09-01289-f007:**
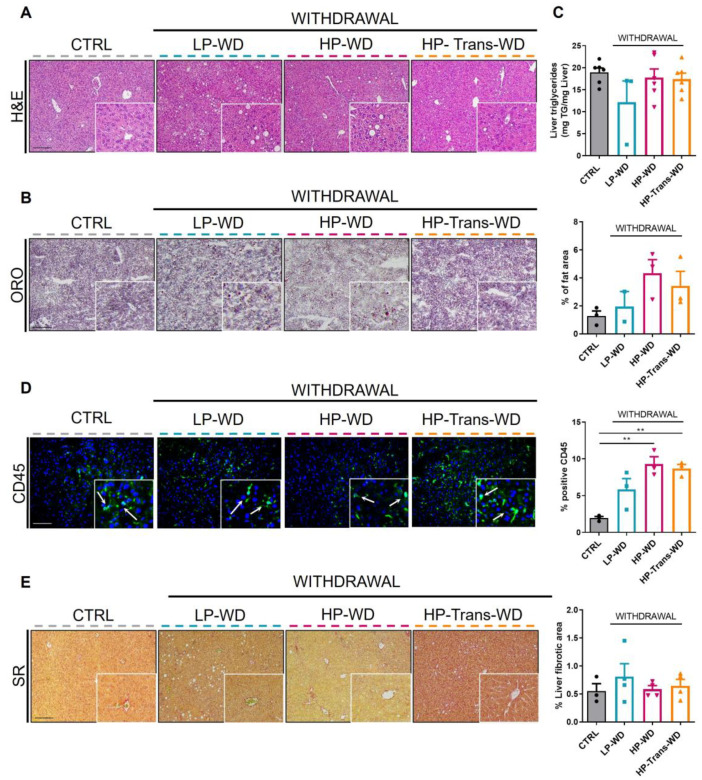
Effects of WD withdrawal on steatohepatitis and fibrosis. (**A**) H and E staining of liver sections from each group. Scale bar = 100 μm (n = 3–6). (**B**) Representative Oil Red O staining of liver cryosections after WD diet withdrawal. Scale bar = 100 μm (n = 3–6). (**C**) Quantification of hepatic TG after diet withdrawal. (**D**) IF staining for CD45. The number of CD45 positive cells (green, arrows) was quantified and calculated as percentage of total cells (DAPI, blue). Scale bar = 100 μm (n = 3). (**E**) SR staining and quantification of positive areas after withdrawal of WD. Scale bar = 100 μm (n = 3–6). ** *p* < 0.01.

## Data Availability

Data sharing is not applicable to this article.

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
