# Peer review of "Fat: Quality, or Quantity? What Matters Most for the Progression of Metabolic Associated Fatty Liver Disease (MAFLD)"

_biomedicines, 2021, doi:10.3390/biomedicines9101289_

Round 1

Reviewer 1 Report

Estevez-Vazques and collaborators examine if qualitative aspects of diet such as levels of PA and the fat source are risk factors for MS and MAFLD. The authors observe that HP and HP-Trans-WD play a crucial role in the genesis of MS and MAFLD. The manuscript entitled “Fat: quality, or quantity? What matters most for the progression of Metabolic Associated Fatty Liver Disease (MAFLD)” is a good study, scientifically valid, well executed, and deserve some space in the journal. However, some concerns have been raised. My major concern is that details of methods in Material and Methods section should be provided:

  • Details of the Immunofluorescence (IF)staining technique should be provided
  • Details of the Immunohistochemistry (IHC) staining should be provided
  • Details of the TUNEL should be provided

Author Response

We would like to thank the Reviewer for the positive comments. In the revised version of the manuscript, we now include the detail description of all above methods in the Supplementary materials and methods section.

Reviewer 2 Report

In the manuscript by Estevez-Vaszquez et al., the authors used mouse Wester diet models to investigate the potential harmful effect of palm oil and trans-fat toward metabolic syndrome and metabolic-associated fatty liver disease. They found that Western diet containing palm oil and trans-fat exacerbated hepatic inflammation and fibrosis, thereby raising caution against the consumption of ultra-processed foods. Experiments were performed in a scientifically sound manner, and results were clearly presented. One caveat is that a similar paper was previously published by Boland et al. in 2019 (Towards a standard diet-induced and biopsy-confirmed model of non-alcoholic steatohepatitis: impact of dietary fat source, World J Gastroenterol 25(33):4904-4920). However, I think that this manuscript contains several new scientific findings and is worth publishing. Therefore, I would like to recommend this manuscript for the publication in Biomedicines after the consideration of following points.

MINOR POINTS

Please refer to the above paper and compare their results with those presented in this work.

Introduction section is rather short, whereas Discussion section is very long. The first two paragraphs of Discussion may be better presented in Introduction. Please check the balance of two sections.

Author Response

Thank you very much for pointing out this paper. The corresponding Reference (49) has been added and discussed in the Discussion section of the revised version of the manuscript (lines 553-555).

Reviewer 3 Report

Well designed, good data, well discussed. Highly recommend to accept.

Author Response

Thank you very much for such a positive feedback.

Reviewer 4 Report

The authors here compared fat source that drives metabolic syndrome and  metabolic associated fatty liver disease. Although some of the results are interesting, conclusion is not supported by current data.

1. Concise method should be provided.

2. It was well established that HFD itself failed to induced heapatitis or fibrosis. Indeed, SR and collagen was stained in the blood vessel but not sinusoid.

3. Mice number was not different from experiment. How many mice were used for induce MAFLD per group and did choose mice for each experiment

4. Fig 4, The authors should provide densitometrical analysis for western blot.

Round 2

Reviewer 4 Report

No more comments